# OpenReview forum: "Simple LLM Baselines are Competitive for Model Diffing"
_ICLR.cc/2026/Workshop/Sci4DL — Sci4DL 2026_

### Official Review · Reviewer_2CNF · 2026-02-24

**Fit:** 3
**Significance:** 2
**Confidence:** 2

**Summary:**

The paper addresses the emerging problem of model diffing, automatically identifying behavioral differences between two language models, especially across revisions or finetuning changes. The authors note that traditional evaluation benchmarks fail to capture unexpected behavioral shifts, motivating automated diffing methods that operate using only API access. They propose a principled evaluation framework based on three desiderata: generalization, interestingness, and abstraction level, and operationalize these using metrics such as frequency, accuracy, abstraction ratings, Interestingness, and acceptance rate. Using WildChat prompts and three experimental settings, including misaligned finetuning, hidden gender assumptions, and frontier model revisions, the paper compares LLM-based diffing with SAE-based diffing. The results show that both approaches achieve similar performance in accuracy and interestingness, but LLM-based methods produce more abstract and actionable hypotheses and achieve higher acceptance rates. The authors conclude that simple LLM-based approaches remain highly competitive and practical for model diffing, especially when internal model access is unavailable.

**Strengths:**

One of the strongest contributions of this paper is the introduction of a clear and well-motivated evaluation framework for model diffing. The identification of generalization, interestingness, and abstraction level as core desiderata reflects a deep understanding of what makes behavioral hypotheses practically useful rather than merely statistically detectable. The operationalization of these properties through held-out validation, LLM-based autorating, and acceptance metrics represents a meaningful step toward standardizing evaluation in this underexplored but increasingly important area. In particular, evaluating hypotheses on held-out prompts avoids overfitting to discovery data and aligns the evaluation with generalization, the core objective of diffing.

A second major strength is the empirical finding that simple LLM-based diffing methods remain highly competitive with SAE-based approaches. This result is both surprising and impactful, as it challenges the implicit assumption in recent mechanistic interpretability literature that internal-representation-based approaches inherently provide superior behavioral insight. The paper convincingly demonstrates that LLM-based diffing can identify meaningful behavioral shifts such as response length differences, stylistic changes, and structural formatting variations, sometimes capturing higher-level behavioral abstractions that SAE features fail to express directly. This has strong practical implications because API-only methods are broadly applicable to proprietary models where internal access is impossible.

The experimental design also includes thoughtful use of model organisms with known ground truth behavioral modifications, which provides a semi-controlled test of diffing effectiveness. Additionally, the inclusion of real-world frontier model revisions strengthens the practical relevance of the findings and demonstrates applicability beyond synthetic benchmarks.

**Suggestions:**

A major limitation of the paper is its reliance on LLM-based judges and autoraters to evaluate hypothesis correctness, interestingness, and abstraction level, which creates a circular evaluation setup where LLMs are both generating and validating behavioral differences. This introduces potential bias, as the judge model may favor hypotheses expressed in familiar linguistic patterns rather than objectively verifying behavioral validity. Additionally, the paper does not report inter-judge agreement, calibration against human annotations, or robustness across different judge architectures, which weakens confidence in the reliability of the evaluation.

This could be improved by incorporating a hybrid evaluation protocol where a subset of hypotheses is verified by human annotators to establish calibration. The authors could also evaluate robustness by comparing judgments across multiple independent LLM judges and report agreement metrics. Complementing judge-based evaluation with objective behavioral metrics such as factual accuracy, hallucination rate, refusal rate, or reasoning correctness would further strengthen evaluation credibility.

Another limitation is that the experimental evaluation primarily captures surface-level behavioral differences such as formatting, tone, and response structure, while deeper cognitive or behavioral differences such as reasoning quality, hallucination tendencies, factual reliability, and safety boundary shifts remain largely unexplored. This limits the strength of the conclusions, as detecting shallow stylistic differences is significantly easier than detecting deeper semantic or decision-level differences that are more relevant for safety and deployment.

The authors could address this by evaluating diffing performance on structured reasoning and factuality benchmarks such as GSM8K, MATH, or TruthfulQA, and by measuring whether the diffing methods can detect known differences in reasoning accuracy or hallucination rates. Using targeted behavioral probes and counterfactual prompts designed to stress specific capabilities would also help evaluate whether the framework can capture deeper behavioral divergence.

The comparison with the SAE baseline is also somewhat limited, as sparse autoencoder performance is highly sensitive to design choices such as sparsity strength, feature dimensionality, training configuration, and layer selection. The paper does not provide sufficient evidence that the SAE baseline was optimized for this task, making it unclear whether the observed performance reflects intrinsic limitations of SAE-based diffing or suboptimal configuration.

This could be improved by performing ablations across SAE hyperparameters and reporting performance sensitivity. Additionally, including alternative representation-level methods such as probing classifiers, representation similarity analysis, or embedding-based behavioral clustering would provide a more comprehensive interpretability baseline and strengthen the comparison.

Finally, the use of WildChat as the primary prompt distribution may bias the discovered differences toward conversational and stylistic behaviors while underrepresenting domain-specific behavioral differences such as reasoning failures, hallucinations, or safety violations. Since diffing effectiveness fundamentally depends on prompt coverage, this limits the generality of the conclusions.

The authors could improve this by incorporating prompts from diverse behavioral domains including reasoning, factuality, safety, and coding tasks. They could also measure how diffing performance varies as a function of prompt diversity, which would provide insight into the robustness and completeness of API-only diffing approaches.

---

### Official Review · Reviewer_J58x · 2026-02-24

**Fit:** 2
**Significance:** 2
**Confidence:** 2

**Summary:**

Given access to model completions to a diverse set of prompts, the authors evaluate two methods to generate hypotheses using SAE features or an LLM based summarization of completions. They use LLM judge based evaluation methods for these hypotheses like interestingness, abstraction level, generalization on unseen examples, and LLM judge acceptance rate.

**Strengths:**

The authors compare two natural approaches to generating hypotheses for understanding features and formalize reasonable evaluation metrics.

**Suggestions:**

The authors only come up with and test evals relying on an LLM judge (I believe all Gemini-Flash). It would be interesting how different judges perform / whether they always agree.

The findings are not that surprising. For instance with an LLM judge evaluating hypotheses from an LLM that the acceptance rate is higher for the LLM outputs. Maybe trying more ways of getting model features like predictive concept decoders etc. would be helpful. SAE emphasis feels quite similar to the Jiang paper and I'm not sure that the metrics / pure LLM-based diffing are sufficient for a full length work if that's the goal. Also exploring faithfulness like topics in this framework could be interesting.

---

### Official Review · Reviewer_d8ZS · 2026-02-25

**Fit:** 2
**Significance:** 3
**Confidence:** 2

**Summary:**

The authors propose a systematic evaluation framework for model diffing, a technique used to automatically surface behavioral differences between language models that traditional evaluations might miss. They identify three key desiderata: generalization, interestingness, and abstraction level, and implement them into metrics to compare existing API-only diffing methods.

**Strengths:**

- **Reasonable and Comprehensive Desiderata:** The proposed criteria for evaluating model diffing methods are relevant and well-justified, including:
    1. Generalization: Assessed via frequency and accuracy on held-out data to ensure the discovered patterns are not mere artifacts
    2. Interestingness: Evaluates whether the method surfaces novel or surprising differences rather than trivial formatting shifts.
    3. Appropriate Abstraction: Measures whether the hypothesis is useful, ensuring it is neither too specific to apply to broad data nor too generic to be discriminative.
- **Insightful Methodological Comparison:** The finding that LLMs offer a superior level of abstraction over SAEs is interesting, and makes sense. I think this ability to capture nontrivial behavioral differences could be useful to practitioners.

**Suggestions:**

- **Expansion of Test Settings**: Future work should involve testing on a larger volume of data and a wider variety of settings and models to further validate these findings.
- **Extrinsic Evaluations**: While the intrinsic metrics are strong, could the authors imagine more downstream or "extrinsic" evaluations for model diffing? For example, assessing the effectiveness of these methods in a safety audit context to specifically detect emergent or hidden harmful behaviors.
- **Human-in-the-Loop Validation**: While LLM-as-a-judge results are promising, the authors acknowledge the inherent unreliability and inconsistencies of automated judges. Integrating human expert evaluation will be important to take this beyond the preliminary stage.

---

### Meta-Review · Area_Chair_3VvU · 2026-03-01

**Recommendation:** Accept

**Metareview:**

Strong fit for the workshop, fair contributions. Recommending acceptance.

---

### Decision · Program_Chairs · 2026-03-02

Accept